# Poor Welfare Indicators and Husbandry Practices at Lion (*Panthera Leo*) “Cub-Petting” Facilities: Evidence from Public YouTube Videos

**DOI:** 10.3390/ani12202767

**Published:** 2022-10-14

**Authors:** Saryn Chorney, Alyssa DeFalco, Jennifer Jacquet, Claire LaFrance, Melanie Lary, Hildegard Pirker, Becca Franks

**Affiliations:** 1Department of Environmental Studies, New York University, New York, NY 10003, USA; 2FOUR PAWS International, Head of Communications, Research and Campaigns Officer, Four Paws International, 36 Bromfield St, #410, Boston, MA 02108, USA; 3LIONSROCK, Head of Animal Welfare Department, Sustainable Sanctuaries, LIONSROCK Big Cat Sanctuary, P.O. Box 1416, Bethlehem 9700, South Africa

**Keywords:** lion protection, compassionate conservation, wildlife tourism, animal welfare, human–animal interactions, social media, content analysis, species-specific behavior, wildlife exploitation

## Abstract

**Simple Summary:**

The present study documents animal welfare concerns associated with a popular form of wildlife tourism: lion cub “petting”. With data collected from 49 publicly available videos uploaded by tourists in South Africa, we find that lion cubs in these facilities regularly exhibit stress behaviors, including stereotypies, during tourist interactions. Video analysis shows that cubs in these facilities are subjected to several forms of poor husbandry practices such as being forced to interact outside normal waking hours. These results also suggest that regardless of tourism type—pure entertainment (tourism) or educational or pseudo-training opportunities in animal husbandry for volunteers (voluntourism)—“cub-petting” facilities involve acute negative welfare impacts in addition to the broader harms of normalizing the touching of young lions and perpetuating captive breeding.

**Abstract:**

There is growing concern about captive lion hunting and breeding operations in South Africa, including cub-petting tourism. For the first time, we assess the quality of cub-petting facilities and code the stress behaviors of lion cubs when handled by tourists by analyzing four stress-related behaviors and six indicators of poor husbandry in 49 YouTube videos of tourist–lion cub interactions (from at least 11 South African safari parks, 2008-2019). We also categorized videos as regular tourism vs. voluntourism (tourism under the guise of helping those in need). We found a median of four poor husbandry practices per video, with all but two videos involving very young cubs (under 7 months) and the majority (61%) involving cubs estimated under 3 months old. Two videos claimed to show cubs as young as 9 days old and 1 day old, with their eyes still closed. The lion mother was apparent in only 1 of 49 videos. All but one of the interactions took place during the day, although young cubs are primarily active at night. The majority of videos (77%) showed cubs engaging in at least one stress behavior, and the most common stress behaviors were avoidance and aggression. Comparing voluntourism to regular tourism, we found no difference in instances of poor husbandry or observable stress behaviors (p’s > 0.6). These results show that cub-petting operations are characterized by poor cub welfare, including features that are inherently harmful to cub development (e.g., separation from the mother at an early age and forced activity outside normal waking hours). Existing research suggests that many tourists are likely unaware of these negative impacts and may even believe that they are helping the cubs. This analysis provides evidence to the contrary.

## 1. Introduction

Although lions (*Panthera leo*) once roamed Europe, the Middle East, India, and Africa, they now live on less than 25% of their original historical range [1] and are classified as vulnerable by the IUCN Red List. In just the past two decades, the total global wild lion population has plummeted from 75,000 to 30,000 individuals [2]. While their wild populations dwindle, captive breeding has increased, and some estimates suggest that more lions now live in captivity than in the wild [3]. Captive lion welfare is thus an urgent concern for the individual lions involved and bears implications for the species as a whole. 

As one of the most sought-out animals in the world, lions are thought to generate significant revenue for zoos, parks, and local communities, especially as lions are one of the “Big Five” species that top both tourists’ and trophy hunters’ lists [4]. In eastern and southern Africa in particular, many countries have developed infrastructure to support lion tourism, with a variety of establishments and vendors focusing on experiences between people and captive lions. This genre of activities, which are often referred to as wildlife tourism attractions (WTAs), may include animal–visitor interactions (AVIs). Internationally, direct contact with captive wild animals—which may include petting, hand feeding, riding, and walking or swimming in their enclosures—is most commonly advertised and may potentially impact animal welfare [5]. 

Lion tourism involves the breeding and rearing of cubs in commercial farming facilities where the animals are used for multiple purposes. Recent estimates indicate that at least 6000 to 8500 captive lions [6] are held in 300–400 or more breeding farms [7] across South Africa alone, though the true number is likely to be higher (e.g., 10,000) as cub-petting operations were not monitored or subjected to government oversight [5,6,7] until very recently. Big cats bred under these circumstances typically become a part of the “canned-hunting” industry, where they are released, often under sedation, into fenced areas to be shot by trophy-hunting tourists. In May 2021, the government announced it would begin to enact measures to limit the lion farming industry, such as halting new hunting permits, prohibiting breeding, and putting a stop to tourism interactions with captive lions. However, the timeline for these changes is unclear [8]. As such, further documentation of the industry is needed to demonstrate the urgency of the situation. 

One area of lion welfare that has received less scrutiny or scientific attention is “cub-petting” facilities, also referred to as “safari parks” or “lion walking” facilities, and more generally as “WTAs” (wildlife tourist attractions) or captive-wildlife tourism—viewing or interacting with wild animals in humanmade confinement, principally zoos, wildlife parks, animal sanctuaries, aquaria, circuses, and shows by mobile wildlife exhibitors [9]. In these operations, tourists pay for the opportunity to have hands-on experiences with cubs, sometimes under the guise of volunteer or “voluntourist” programs that sometimes promote the false pretense that the cubs are orphans and that the facility is a sanctuary [10]. Moorhouse et al. report that “only a minority of tourists was alert to welfare conditions at WTAs” and “the majority of tourists do not perceive poor animal welfare conditions” [9]. Whether tourists are lacking in education and the ability to recognize poor animal welfare or if they ignore certain welfare factors so as not to detract from the overall interactive experience is unknown, though evidence suggests that tourists do respond negatively to conspicuous abuses and compromised welfare standards for certain captive species [11]. Though the details surrounding these facilities—where the cubs originate, how they are treated and housed, and what happens to them after they become too large to interact with tourists—are somewhat murky, the existing evidence points to welfare concerns throughout the life cycle. 

In contrast to claims that the cubs in these facilities are part of rehabilitation programs, it seems likely that they are the product of deliberate breeding programs and destined for canned-hunting outfits [12], the exotic pet trade, or the growing lion skeleton trade [13]. Cubs born into captive farms, petting zoos, and complicit lodges are often the first phase in a “sprawling, vertically integrated captive-breeding/canned-hunting commodity chain, which caters to thousands of foreign hunting clients annually” [12]. As such, the ultimate destination of lions from cub-petting facilities typically involves conditions with extremely poor welfare. Even less is known, however, about the welfare of the cubs during their time at the cub-petting facilities themselves.

To address the question of the welfare of cubs while they remain at the cub-petting facilities, we analyzed tourist videos posted on social media. YouTube studies are becoming more popular, employing publicly available video data to document, analyze, and gain information about animal behavior, animal welfare, husbandry practices, and captive living conditions that are difficult to otherwise record. As Pokharel et al. (2022) state, “detailed description … requires the observer to be at the right place at the right time”, and despite the varied video durations and shortcomings in terms of more traditional data collection methods, social media can provide useful insights into “iEcology” and “promote public engagement in science” [14]. As with other forms of tourism, “selfie” culture abounds in cub petting, providing a sample of publicly available video clips taken at cub-petting facilities. Recent work has suggested that social media provides a unique source of animal behavior and welfare data, especially in cases where first-hand data collection may be difficult to impossible [15,16]. Taking advantage of this material on YouTube, we generated a database of potentially usable clips by searching publicly available videos and restricted our search to South Africa, where most of the facilities operate. After sorting the relevant videos, we coded the material according to several welfare-relevant husbandry parameters (e.g., time of day, cub age) and stress behaviors (e.g., avoidance, aggression). Analyzing these video data provided a preliminary assessment of the welfare of cubs in cub-petting facilities that may be of use for future lion welfare and protection work. The goal of this study was to shed light on the cub-petting industry, as well as to provide some conclusions about cub petting for future research on welfare implications in relation to tourist activities and public/social media attitudes. 

## 2. Material and Methods

### 2.1. Ethics Statement

At the time of coding, all videos used in this research were published on YouTube with the standard YouTube license or Creative Commons CC BY license, both of which allow for viewing without restriction. Behaviors were coded by authors S.C. and A.D. No identifying information was obtained, and no animals were harmed or manipulated for the purposes of this study. 

### 2.2. Generating the Video Database

We accumulated clips using the search phrases “South Africa”, “lion cubs”, “petting”, “baby lion”, “cub”, “petting”, “playing with lion cubs”, “cheetah experience” (lion cubs and cheetah cubs are frequently housed together), “Zorgfontein”, “Horseback Africa”, “Lion and Safari Park”, and “ Ukutula”. Many specific South African locations used in search terms were previously identified as having cub-petting experiences from the 2018 book *Cuddle Me, Kill Me* by Richard Peirce and were verified through the establishments’ websites. Other specific locations were identified via the YouTube videos found through previous non-location-specific search terms and again verified by checking the facilities’ websites for cub-petting advertisements. Observations are based on footage taken at known and suspected captive breeding facilities, safari parks, petting zoos, complicit lodges, and other animal entertainment attractions in South Africa.

We excluded videos containing non-petting and non-handling lion cub footage. We also excluded videos of other cubs (e.g., bears) and different big cat cubs, such as tigers and cheetahs. Exposé videos about the industry and news videos were also excluded, as were videos that featured adult lions only and older juveniles (> 10 months), as verified by the head of animal welfare at a big cat sanctuary who has been responsible for the care of over 100 big cats, including many lions, for over 13 years.

This procedure led to 267 videos found on YouTube, of which 49 videos were applicable to our study. The videos had been uploaded by users onto YouTube from 2008 to 2019 (median length [range]: 2.75 [0.5, 25.57] minutes). Unlike an exhaustive, intensive sampling procedure (e.g., conducting live scan sampling throughout the day) at one or several of these sites, by providing a more broad, serendipitous survey of activities taking place at “cub-petting” visits across South Africa, these data represent a subset of the total frequencies and a subset of the complete behavioral range of captive lion cubs and husbandry practices. As such, the results should be interpreted as minimum floor estimates of the rates of occurrence and types of husbandry practices occurring during “cub-petting” visits in the area. 

### 2.3. Video Coding

After building and refining the database, two researchers coded each clip according to the variables contained in Table 1, which consists of a variety of environmental conditions and lion behaviors. Variables were selected based on previous behavioral studies of African lions [17,18,19], as well as an enrichment-focused study that also assessed behavior [20]. The reliability between coders was acceptable across all variables: all continuous variable ICCs > 0.92; all categorical variable agreement values > 0.90 and kappas > 0.67. A lion behavior expert at a big cat sanctuary in South Africa and an executive from an international animal welfare nonprofit organization evaluated the videos and our coding methodology for accuracy and reliability in classifying lion behavior and other welfare-related assessments.

### 2.4. Statistical Analyses

We used R [28] and Rstudio [29] to process, visualize, and summarize the data. To assess differences between regular tourism videos and voluntourism videos, we ran non-parametric Kruskal–Wallis tests on (i) the tally of poor husbandry practices evident in each video and (ii) the tally of the stress behaviors that occurred at least once in each video. To assess patterns over time, we ran linear least squares models on aggregated variables (overall tally of poor husbandry practices per year and overall tally of stress behaviors per year). 

## 3. Results

We first examined six indicators of poor husbandry: daytime (under natural conditions, lion cubs are only out of the den, if ever, at night); away from mother (which is unobserved in nature before 3 months of age) [21]; premature age (either less than 7 months or less than 3 months); white lion (an abnormal phenotype popular with trophy hunters); barren housing/lack of enrichment; social isolation (see Table 1 for definitions). Nearly all the videos in our database were filmed during the day (48/49) and without signs of the mother in the vicinity (48/49), and nearly all cubs were estimated to be under 7 months (47/49) and many under 3 months (30/49; Figure 1). Two videos, according to their accompanying descriptions, showed cubs as young as 9 days old and 1 day old interacting with tourists (cubs’ eyes were still closed shut and/or they could not yet steadily walk without falling). In addition, 41% (20/49) appeared to be white lions. The conditions at the facilities were barren, without signs of any environmental enrichment, 49% of the time (24/49), and cubs appeared to be socially isolated from other members of their species in 12% of the videos (6/49; Figure 1). Overall, our database contained videos with a median of four poor husbandry practices per video, with no video showing less than two and several showing at least five.

We next examined four stress-related behaviors: active avoidance; aggression; abnormal behaviors (including stereotypical behaviors); and attempts to hide (see Table 1 for definitions). Seventy-seven percent (77%; 38/49) of the videos contained at least one stress behavior, with the most frequent behaviors being avoidance (26/49) and aggression (24/49), followed by abnormal behavior (10/49) and one video in which the cub was observed actively trying to hide (Figure 1).

### 3.1. Voluntourism

Approximately 20% of the videos (10/49) appeared to take place at voluntourism facilities (see Section 1 and Table 1 for definition). Compared to regular tourism operations, voluntourism involved the same frequency/video of poor husbandry practices (regular: 3.56 vs. voluntourism: 3.70, Kruskal–Wallis: *p* > 0.6) and stress behaviors (regular: 1.23 vs. voluntourism: 1.30, Kruskal–Wallis: *p* > 0.8; Figure 2).

### 3.2. Over Time

During the study period, videos from 2008 to 2019, husbandry practices appeared to improve slightly, with fewer poor husbandry practices evident in videos posted in recent years compared to ten years ago (effect: 0.10 fewer poor husbandry practices per year; 95% confidence interval: −0.22, −0.01; t(47) = 2.12; *p* < 0.04; Figure 3a). While we did not have enough statistical power to test each practice on its own, visually examining the data reveals that all poor husbandry practices tended to decrease over the study period, with barren enclosures showing the most robust decrease. 

In contrast to the housing conditions, stress behaviors became *more* apparent during the study period, with more types of stress behaviors being shown in videos posted in recent years compared to ten years ago (effect: 0.12 more stress behaviors per year; 95% confidence interval: 0.02, 0.22; t(47) = 2.24; *p* < 0.02; Figure 4a). While we did not have enough statistical power to test each behavior on its own, visually examining the data reveals that all stress behaviors appeared more often over the study period, with aggression and abnormal behavior showing the most robust increases. 

## 4. Discussion

A recent survey of lion stakeholders identified the top welfare concerns to be the cubs’ ability to choose their social groups, their own environments, and the choice to retreat from human interaction, as well as the lack of dedicated, trained caregivers and overall poor breeding practices [23]. Evidence of these potential welfare concerns is seen throughout our video database. Out of a possible six indicators of poor husbandry, we found that at least two poor husbandry practices were present in all cub-petting videos with a median of four indicators/video. Out of a possible four stress-related behaviors, 77% of the videos contained at least one stress behavior, with avoidance and aggression being the most common. Voluntourism videos displayed equivalently problematical welfare issues as regular tourism videos. Although husbandry practices appeared to improve slightly over the study period, stress behaviors were found to increase over time, a divergence indicating that simply improving husbandry practices at cub-petting facilities does not inevitably alleviate cub welfare concerns—the core harm may simply be the nature of the “cub-petting” activity itself. 

### 4.1. Abnormal Behavior, Aggression, and Stress

Poor welfare in lions can manifest in a variety of behaviors and is influenced by various social and environmental factors. Captive lions may, for example, pace inside their enclosures and display abnormal or stereotypical behavior stemming from unwanted proximity to another animal, conspecific or human; insufficient feeding; or insufficiently sized enclosures [25]. Pacing may also manifest as a method of releasing frustration or a psychological scar from a past trauma that has developed into an established routine [26]. Nearly 20% of the cub-petting videos contained evidence of such negative responses. 

We also coded for hiding (and locomoting away) from and aggression towards humans, which are additional signs of stress in cubs [27]. The majority of videos contained cub avoidance behavior and nearly half contained aggression. Avoidance behavior is easily recognizable and considered a sign of fear [19]. The lion stakeholder survey identified cubs’ inability to choose to retreat from constant forced interactions as a leading welfare concern, along with the amount of time humans spent interacting with the cubs. Both may lead to an immediate negative behavioral response, as well as a lasting welfare impact regarding future interactions and related long-term problems [23].

Captive-bred lions are farmed and, much like domestic farm animals, handled by a variety of different humans, primarily during the daytime. Lions are largely crepuscular and nocturnal, meaning they are most active at dusk and dawn and more likely to rest or sleep mid-day [23]. Generally, large predators—big cats included—sleep for long periods and appear to sleep deeply [30]. Studies in a variety of mammals suggest that sleep, though species-specific, is a beneficial adaptive state that increases activity efficiency, saves energy, reduces risk of injury, and is associated with processes such as neurogenesis and immune system regulation in mammals [31]. Lion cubs being kept awake and encouraged to be active during morning and daytime petting sessions is therefore a noteworthy concern. Sleep time is “believed to be compromised by the need for people to interact with the cubs for extended hours”, resulting in a lack of sleep or a lack of quality sleep, which are both welfare concerns [23]. Thus, the basic facts of cub-petting encounters (forced removal from the mother and exposure to humans during the daytime) are likely fundamentally incompatible with cub welfare, regardless of various mitigation attempts to ameliorate the harms within the actual setting.

### 4.2. Social and Physical Environment

In the wild, cubs are typically hidden away by their mothers until they are two months old [21] and begin following their mother at three months old; they are not weaned until about seven months old. In this study, videos showed cubs as young as one day old interacting with tourists. Our results found that nearly 96% of cubs featured in tourist interaction videos were under the age of seven months. In zoos, experts suggest that even trained animal husbandry professionals should stop direct interactions with cubs at approximately three months old, the age when they could potentially become dangerous, but most South African facilities offer cubs that are six months old and even older [23]. Additionally, the young lions utilized for human handling experiences are touted as rejected orphans when they are, in many cases, removed from their mothers at birth and hand-reared. Our results indicate that approximately 98% of the videos show no evidence of a mother lion, or even other adult lions, present with the cubs. This practice is problematic for both the cub and the mother. When cubs are taken from their mothers, they are often hand-reared alongside litters of other ages which presents a concern in that older cubs may dominate younger ones without adult cat supervision [32]. Due to the nature of our opportunistic public video research, to some extent we are reliant on the accuracy of the video descriptions in terms of cub age. We acknowledge this limitation and the possibility that some age estimates may not be precise, though we trust that our lion behaviorist co-author has accurately gauged the cub ages to roughly a month. Nonetheless, the nature of the results still indicates the very young age at which these lions are being handled by unexperienced members of the public. 

Removing cubs within days from their mothers is widely understood to be detrimental to young lion health. Both the age of removal and the impact of removal are notable welfare concerns, according to surveyed lion experts [23]. Age of removal is specifically linked to whether cubs had access to colostrum and mother’s milk for an adequate time to gain nutritional benefits, while the impact of removal is believed to potentially cause psycho-physiological stress [23]. Orphan cubs who have not had the benefit of imprinting upon their mother or another conspecific mother figure soon after birth would miss out on a unique form of acquired learning, including the performance of “behaviors at appropriate levels through the developmental stages of cub to adult” and the ability to visually explore their environment [24]. Research has suggested that regularly removing cubs from their mothers (as happens in cub-petting facilities) may increase the possibility of the mother lions eventually rejecting their cubs, raising yet another welfare concern [23].

In general, litters depend on their mother and other female pride members for food and care until they are at least 10 months old [33]. Cubs who are separated from their mothers are often given an alternative milk formula at lion farms and other captive petting facilities which may lead to nutritional deficiencies that weaken their immune system, rendering them vulnerable to pathogens [34]. Additionally, cubs show a strong tendency to remain at their mother and pride lionesses’ sides through maturity, about two to three years in the wild [35]. African lion prides tend to have a consistent social structure, including the need for a learned dominance hierarchy. A 2016 study on the subjective well-being and personality of big cats, including African lions, indicated that elements of lion dominance and social structure are particularly important aspects of welfare [36]. A cub’s “exposure to adult lions is a critical component” of social development and all options, including males, should be considered [23].

The mother lions are likely to suffer from the separation as well. While the psychological effects on the mother cat have not been sufficiently studied, surveyed lion experts report mother cats potentially suffer separation anxiety which is exacerbated by repeated forced removals [23]. Anecdotal evidence, such as scenes depicted in the documentary *Blood Lions* [37], show cubs and the mother cat crying for each other upon separation and suffering in myriad emotional (e.g., stress vocalizations) and physical (e.g., bone deformations and more genetic issues due to inbreeding and crossbreeding) ways. White lions only occur naturally in one African nature reserve and adjoining park, yet they are commonly found in captive lion breeding facilities throughout South Africa, indicating they are being severely inbred, which leads to many defects and susceptibility to disease [23]. As such, cub-petting facilities may entail harm to adult lions in addition to the cub welfare issues documented in this research. 

The spatial and structural details of the environment are also likely to affect lion welfare. According to a 2000–2001 study conducted in the U.K., African lions living in captivity had around 3.4 hectares (8.4 acres) of enclosure space in comparison to a minimum of 672.9 hectares (1662.77 acres) of home range [29]. Such drastic reductions in spatial availability are known to be associated with indicators of poor welfare in lions [38]. Moreover, research has shown that Asiatic lions prefer habitats that mimic their home range [35], and it is likely that African lions have similar preferences for habitats that mimic the savanna or grassland environments of their home range. One of the six indicators of poor husbandry we looked at in this study was barren housing and lack of enrichment in enclosures. Nearly half the enclosures viewed in the cub-petting facilities were barren, without signs of enrichment added to the environment. Lion caretakers suggest that “more space and choices about where lions can spend their time may prevent social and behavioral problems” [23]. The combined lack of adequate space, spatial complexity, and environmental enrichment offered in the cub-petting facilities contributes to additional welfare risks.

### 4.3. Implications of Animal Tourism

The terms “orphan” and “sanctuary” are both regularly used by cub-petting facilities, but both descriptions are likely misleading and may be used for advertising purposes [39]. To correct these misperceptions, an educational awareness campaign could help to inform the public about the unnatural circumstances and conditions linked to facilities that use animals for profit. Additionally, both the industry itself and the tourists who support it may benefit from increased transparency and more regulation regarding various forms of greenwashing, including misleading language and the kinds of facilities that may call themselves sanctuaries. Sanctuaries uphold privacy for the animals, including the relative quiet of life in nature, relaxation from close confinement and social stress, and “environments or vistas that expand their visual, auditory, and olfactory sensations” [39]. The results presented here show that elements of sanctuaries are not present in facilities that offer opportunities to “pet cubs”. While sanctuaries provide animal husbandry and veterinary care, they do not seek to manipulate the animals’ behavior on behalf of human desires. Even though numerous studies indicate that forced proximity to humans is a source of stress for wild animals [40], cub-petting facilities promote human interaction with lions. Lack of experience, lack of proper volunteer training, and inappropriate handler behavior may cause fear of humans, yet another long-term welfare concern, as well as health issues related to daily human contact [23]. In contrast to the advertised notion that the young cats are orphans being rehabilitated for the purpose of reintroduction to the wild, this is the fate of few if any captive cubs [39]. These young lions are ill-suited for release in the wild due to inbreeding, lack of survival skills, and their familiarity with humans. Consistent with this, we found no evidence that voluntourism cub-petting facilities were different from the regular cub-petting facilities: voluntourism involved the same rate of negative husbandry practices and stress behaviors as regular tourism operations. 

According to a 2012 Australian study, animal tourism has a negative net environmental effect. This is especially true for IUCN Red List species; less than 30% of wild African lion populations are protected via tourist-related conservation funds [41]. Captive breeding facilities rarely receive tourism-sponsored conservation funds, yet the practice has widespread support from the general public and local governments [10]. Many former cubs end up being sold to ranches and safari hunting outfits as part of the canned-hunting industry. The animals are exploited by breeders and other players in the lion farming industry for profit. Lion hunts on game ranches in South Africa may cost from USD 20,000 to USD 40,000, and captive-bred lions are often preferred by hunters because their manes and fur are typically in better condition than those of wild lions [5]. Thus, human-reared cubs may be even more financially valuable in adulthood than their wild counterparts. In some cases, as seen on the Cheetau Safaris website’s hunting price list, a male lion can be selected for killing for upwards of USD 10,000 while the female is listed for USD 5500 [38].

Thus, when the cubs from cub-petting facilities grow too large to remain at the petting facility, they continue to transition through a series of “lively commodity” forms [42], which serve various audiences and paying clients, including trophy-hunting outfits, exotic pet dealers, and the growing lion skeleton trade that runs rampant in Southeast Asia and China where legal lion bones are used in tonics or lion bone wine and as a substitute ingredient for tiger parts in traditional Chinese medicines [13]. Lion experts agree the leading welfare concern for lions is a lack of regulation within the cub-petting industry, including ethical concerns about cubs exiting into the canned-hunting industry and/or bone trade once they are too old for petting [23]. Additionally, a portion of the captive female and male lions of breeding age are kept in facilities for a number of years to continue the cycle. 

Due to these widespread and harmful issues, TripAdvisor and its Viator brand have announced a new welfare policy whereby the company will no longer “sell tickets to or generate revenue from experiences where captive wild or endangered animals are forced to perform demeaning tricks or other unnatural behaviors in front of the general public” [43]. Similarly, Airbnb launched a responsible “Animal Experiences” booking platform in conjunction with World Animal Protection which prohibits direct contact with wild animals, including but not limited to petting, feeding, and selfie-taking, specifically banning big cat interactions, illegal wildlife trade, canned and trophy hunting, and animals performing for entertainment [44]. These actions are necessary, because as Moorhouse et al. point out, tourists “are consumers whose primary goal is the consumption of a tourism experience as an escape from their everyday existence [and] may not view [vacations] as a context in which they need to behave in an environmentally responsible manner … or may, when abroad, participate in activities they would normally avoid” [9]. 

A 2020 study performed by South Africa’s Minister of the Department of Forestry, Fisheries and the Environment (DFFE) initiated a high-level panel review on the management, breeding, hunting, trade, and handling of lions, as well as elephants, leopards, and rhinos, in the country. As a result of the panel’s December 2020 conclusions and the Cabinet’s April 2021 approval of these recommendations, the Minister of the DFFE announced plans to end the domestication of lions; halt lion captivity; ban the sale of lion body parts and derived products; and effectively end commercial lion farming, inclusive of breeding, hunting captive-bred lions, and tourism interactions such as cub petting. These recommendations still need to pass through Parliament for approval by the Cabinet; however, the timeline is unknown [8].

As interactions with wildlife relate to the current global pandemic, WTAs in South Africa allow tourists “to be in direct contact with animals, which may lead to the transmission of pathogens from animals to humans” and humans to animals [23]. Unnecessary and inappropriate contact with wild animals, inclusive of tourist attractions, is an aspect of modern life that ought to be left in the past.

### 4.4. Future Lion Protection Work

The present research relied on an “opportunistic” method of data collection, rather than controlled in-person methods. As such, some videos may not have captured the full range of behaviors in a given interaction and the tally of stress behaviors should be considered a floor estimate. Videos posted to YouTube publicly are likely ones deemed acceptable and appropriate (or “cute” and entertaining) by tourists and voluntourists, yet even these interactions show signs of stress and poor welfare. It is possible that many more private videos and recordings never posted online show overtly “bad” or “cruel” interactions and far worse welfare at these types of facilities. Despite these limitations and even as a baseline estimation, the data reveal concerning welfare issues and harms involved in cub petting. This study focused on activities in South Africa as it is a hub of cub petting and captive lion breeding, but further YouTube research could include facilities in other countries around the world where exotic petting zoos are popular (e.g., the United States).

To provide a contrast, more research is needed to examine the potential benefits of wildlife sanctuaries. Sanctuaries may inspire visitors to reconsider the connections between people and wild animals, as well as cause them to rethink their individual responsibility and role in perpetuating captivity in the form of “circuses, zoos, laboratories, marine parks, film productions, cub petting operations, and other exploitative situations” [36]. Future work could investigate whether visitors are more likely to develop an appreciation for animal welfare and environmental protection when they visit facilities that value the inherent worth and dignity of the animals above human education and entertainment. 

## 5. Conclusions

Much of the media and public concern regarding captive African lion welfare is focused on captive hunting. Hunting is only one aspect of a larger commodification life cycle, beginning with cub petting and ending in either trophy hunting or the illicit bone trade. Currently, support from the South African government enables these activities, though legislation could change the status quo for these vulnerable animals. In October 2019 [45], a pre-established committee formed by the South African Parliament resolved to initiate policy and legislative review of captive lion breeding, cub interaction facilities, hunting, and the lion bone trade. The committee’s stated purpose was to protect South Africa’s conservation image and tourism revenue, though the High Court ruling addressed cruelty and other problems within the industry, stating that welfare and conservation were intertwined values. Per the 2021 high-level panel recommendations by the Minister of DFFE to end commercial lion farming in South Africa, it is necessary to examine a variety of issues related to the management of a responsible exit plan. Recent issues identified by a gap analysis include diverse and complex, yet actionable, “steps to aid the process towards a regulated, transparent and well-monitored time bound transition that mitigates potential unintended negative consequences for the lions and people currently operating within the industry” [8]. Regulation, guidelines, lion welfare, conservation, health and safety, and more issues need to be addressed by various stakeholders in order to develop and implement this process. Though rare, cases of zoonotic disease transmission between humans and captive lions in close proximity have been scientifically documented, and generally speaking, high concentrations of captive wild animals may suffer from factors such as poor diet and stress, which are known to cause a higher risk of infectious disease transmission, posing a threat to both the lions and public health [34]. One of the simplest ways to manage and prevent these risks globally is to limit regular or prolonged close contact between humans and wild animals.

In the interim, as tourism rebounds in South Africa, those who seek out immersive animal experiences will likely require a targeted awareness campaign. Both the animated 1995 Disney film and 2019 live-action version of *The Lion King* reiterate the dangers of the “Simba effect”, i.e., tourists who want to hold a lion cub in the air and take photos or videos of the experience; this is an ongoing social media temptation and overall welfare concern. By studying user-generated content in the form of YouTube cub-petting videos, our research provides a closer look at the problems related to human handling and lion cub welfare and may serve to enhance awareness of the negative consequences of lion cub tourism. 

In conclusion, the results of this study suggest that cub-petting activities are likely to have a net negative impact on lions. At a minimum, if captive lion breeding cannot be stopped immediately, transitioning cub-petting facilities into sanctuaries that adhere to principles of non-contact and uphold the inherent worth and dignity of the lions would provide a more enriching opportunity for the humans who visit, volunteer, or work there, as well as maintain a healthier, more species-appropriate environment and safer option for the lions. 

## Figures and Tables

**Figure 1 animals-12-02767-f001:**
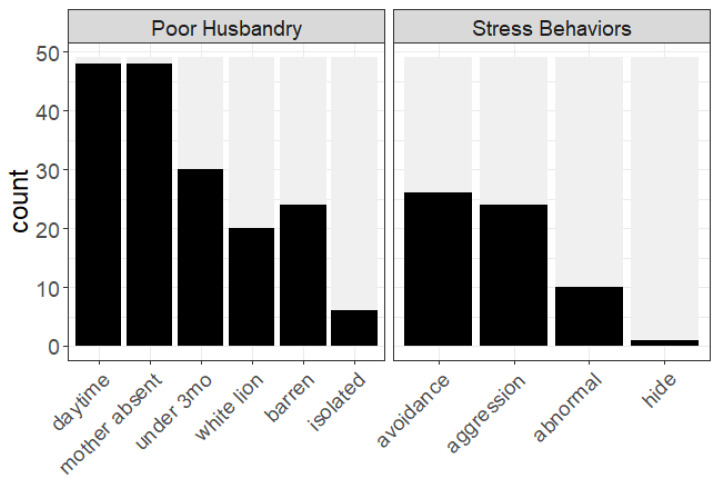
Videos characteristics. The black bars represent the number of videos (*n* = 49) that showed each of six poor husbandry practices and four cub stress behaviors (see Table 1 for definitions).

**Figure 2 animals-12-02767-f002:**
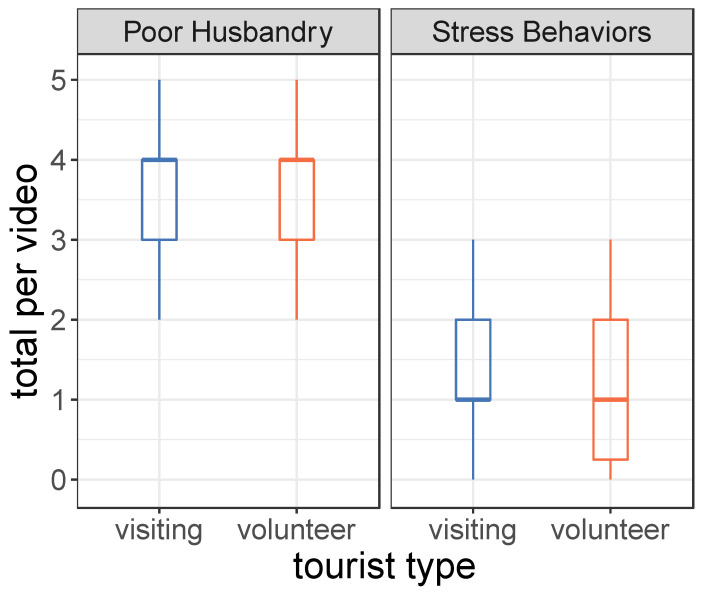
Evidence of poor husbandry and cub stress by tourist type. The thick horizontal lines, colored boxes, and whiskers (standard box-plots) indicate, respectively, the median, central 50% bulk of the data (interquartile range), and the outer percentile of observed poor husbandry practices (6 max) and cub stress behaviors (4 max) in each video (*n* = 49). Kruskal–Wallis tests indicated no difference in these variables by tourist type (both *p*’s > 0.6).

**Figure 3 animals-12-02767-f003:**
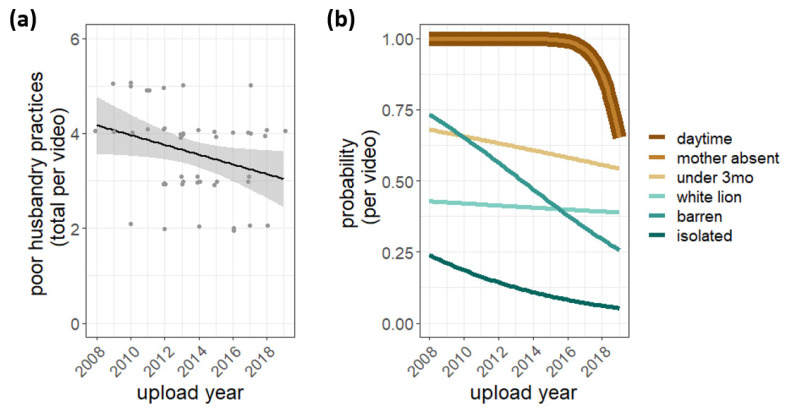
Poor husbandry practices over time. (**a**) The dots are the total number of poor husbandry practices in each video (6 max). The black line and grey band are the best-fit linear least squares line and its associated standard error (*p* < 0.04). (**b**) Each line represents the logistic regression model for the probability of observing a particular poor husbandry practice in a video for each year. Note: Double line at the top of Figure 3b represents the curve for “daytime” and “mother absent”, which followed the identical trajectory. All poor husbandry practices tended to decrease over time, but due to low power, testing for statistical significance of each poor husbandry practice in isolation was not possible.

**Figure 4 animals-12-02767-f004:**
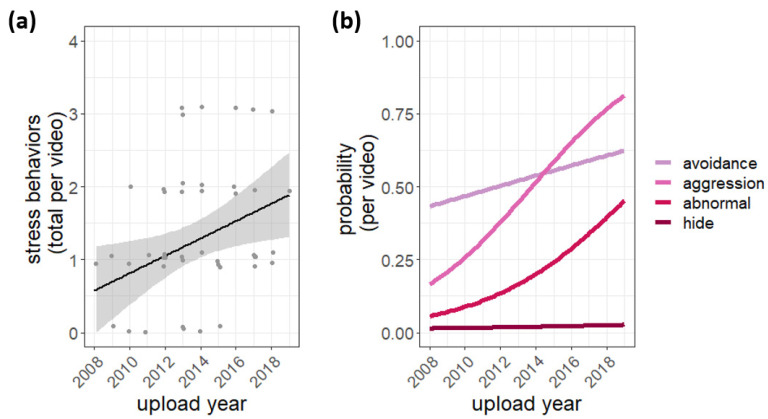
Cub stress behavior over time. (**a**) The dots are the total number of cub stress behaviors in each video (4 max). The black line and grey band are the best-fit linear least squares line and its associated standard error (*p* < 0.02). (**b**) Each line represents the logistic regression model for the probability of observing a particular poor husbandry practice in a video for each year. All stress behaviors tended to increase over time, but due to low power, testing statistical significance for each behavior in isolation was not possible.

**Table 1 animals-12-02767-t001:** Behaviors and conditions; descriptions of lion cubs analyzed.

Categories	Variable	Description	References
Husbandry concerns	Cubs younger than 3 months old present without a lioness	Very young cubs’ eyes are closed or color is gray-blue to amber; coats are woolly, grayish yellow with many dark spots on forehead. In the wild, cubs are hidden by mothers until two months old and begin following their mother at three months old.	[21]
	Cubs under 7 months old present without a lioness	Older cubs have sleek, short-haired tawny coats similar to adults; legs and sides may be faintly spotted and tail tuft is more prominent. Cubs are not weaned by mothers until about seven months old.	[21]
	Barren	The absence of objects such as toys, climbing rocks, and adequate space that adds value to the lion’s experience in the enclosure.	[20,22]
	Daytime	Video was taken during the day, despite lions being crepuscular and/or nocturnal.	[23]
	Isolation	Cub is completely isolated from any other lions.	[23]
	Mother absent	Lack of proximity or interaction with mother or any adult lion pride members.	[21,24]
	White lion	Rare color mutation of African lions.	[23]
Stress behaviors	Abnormal	Auto injury/mutilation, refusal of food, auto feces licking, head tossing, pacing, signs of illness.	[25,26]
	Aggressive	Interaction with other cubs, humans, or enrichment that is threatening or reactive; aggression (vs. play) is characterized by a higher tendency to growl and snarl, often in conjunction with tense bared-teeth and tail whipping.	[21,27]
	Avoidance	Locomotion from one place to another, i.e., walking, running, trying to escape, actively retreating from humans.	[19]
	Hide	Cub avoids humans and/or other cubs by isolating itself from group but does not involve locomotion.	[27]
Tourism types	Voluntourism	Video of facility features individuals wearing uniform t-shirts, who feed and/or bottle-feed, clean pens, and handle cubs. Videos often higher production quality.	[10]

## Data Availability

Data is contained within the article or Appendix A.

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
