# Peer review of "Poor Welfare Indicators and Husbandry Practices at Lion (Panthera Leo) “Cub-Petting” Facilities: Evidence from Public YouTube Videos"

_animals, 2022, doi:10.3390/ani12202767_

Round 1

Reviewer 1 Report (Previous Reviewer 1)

Thank you for incorporating suggested revisions. These changes have resulted in a much improved, more focused paper. I had three small grammar comments and one question.

1. Table 1: For the definition of Barren, I think you mean "absence" not "presence"

2. I'm curious: Why do you think stress behaviors are higher in more recent videos, even with changes to husbandry? Were later videos more prone to filming these behaviors because they were they longer? Granted more access? 

3. The font is weird on lines 329-332.

4. Line 480--where is the paired mate to this quotation mark?

Author Response

Thank you for your comments and the time you took reviewing our MS.  We are very grateful for the resulting improvements to our work.  Please see attached cover letter for more details.  

Reviewer 2 Report (Previous Reviewer 2)

The authors have implemented all suggestions for minor changes I provided in my initial review.

Author Response

Thank you for your comments and the time you took reviewing our MS.  We are very grateful for the resulting improvements to our work.  Please see attached cover letter for more details.  

This manuscript is a resubmission of an earlier submission. The following is a list of the peer review reports and author responses from that submission.

Round 1

Reviewer 1 Report

General comments

The core of this project is comparing conditions and behaviors that are potential negative welfare inputs and outputs at “regular” tourist facilities where lion cubs can be touched to “voluntourism” tourist facilities that offer similar experiences. The work done to find, score, and analyze these factors and responses is reasonable with some additional detail required about the methods. However, the paper does have flaws that require attention. There are multiple places where long quotes are pulled and copied in, and this practice should be avoided if possible. More seriously, the paper suffers from a lack of focus.

A weakness of the paper as submitted is the inclusion of many sections where the focus of the text doesn’t quite match the focus of the methods and results. From the title, I would expect a deep dive into rates of handling or petting, or—as was used in the paper—investigation of the settings in which petting occurs, and then some combination of behavioral and physiological measures to determine the effect that petting or conditions had on cubs. This comparison between welfare inputs and outputs never occurs. Instead, the methods and results investigate the differences (or lack thereof) between “regular” and “volunteerism” facilities—this is a fine topic, but the current title of the paper does not really reflect that content.

Again—as written, the paper isn’t about the intensity or the practice or absence of “cub petting” at these facilities per se but rather the conditions between what are supposed to be two types of tourist facility where cubs are petted and how observations from video reveal an actual lack of difference. However, this specific comparison between the two types of facilities is not the focus of the extended discussion, conclusion, or intro, which each digress into sections about the entirety of the captive lion trade from cubs to canned hunts of adults, and are therefore outside the scope of the work performed by the authors.

Talking about how volunteerism facilities are presented as being different from and more wildlife friendly than regular tourist attractions while being functionally identical and producing the same categories of stress behaviors in lion cubs as “regular” tourism is fair game. If this paper is instead meant as an indictment of the cub petting industry or of volunteerism facilities, you would need for comparison some baseline data from true sanctuaries/accredited facilities or more data-driven examination of the connection between welfare inputs and outputs. I understand that such data are hard to gather, since the (presumably) author-approved facilities don’t offer cub petting and the pre-existing videos used in this project offered only opportunistic data collection. Without those baseline behaviors for comparison, it is difficult to say that the welfare of cubs is “better” in one kind of facility, limiting the discussion to these tourist locations only.

Specific comments

TITLE

As discussed above, the focus of the results of the paper as written isn’t really the impact of cub petting on behavior and welfare, but rather the comparison of husbandry conditions and stress behaviors observed between two types of tourist facilities. Either the title must change or the focus of the paper must.

SIMPLE SUMMARY

Lines 14-17 of the summary are appropriate to the work done but should probably mention the comparisons between facility type.

Line 18-20. Cut from “Cub petting facilities…” to the end of the summary. These lines digress and speculate from your specific performed work.

ABSTRACT

Abstract is fine and is representative of the work performed, no notes.

INTRODUCTION

Line 45. Delete text: “The so-called ‘Lion King’ is on the decline.”

Line 59. Consider restricting/reordering this part of the introduction so that the mention of cub petting on line 59 occurs after its longer explanation that starts on line 66.

Line 60-65. While important from an awareness point of view, canned hunting of adult lions and extended discussions of canned hunting drift from the main focus of the paper (examination of the welfare of cubs in various tourist-driven facilities). Suggest deleting text that starts on lion 60: “Big cats bred under these circumstances…” and ends line 65 with “… is needed to demonstrate the urgency of the situation.”

Line 68. Em dash should not have spaces before and after it

Line 70. Delete “unwitting”

Line 73-77. The quote that starts with “the majority of tourists…” is long and a large chunk of copy-pasted text that goes on for multiple sentences. Please summarize and try to avoid using lengthy sections lifted from other publications.

Line 92. Change “…publically available videos, restricting our search…” to “publically available videos and restricted…”

MATERIALS AND METHODS

Line 111-112. How are “real sanctuaries, rescue centers, and wildlife reserves” defined? Are “accredited zoos” accredited through PAAZA or another accrediting body? From a design point of view, could you have obtained as a baseline any footage from “real sanctuaries?” Although behaviors during cub petting would not have been observable from these “real sanctuaries”, could you have looked at behaviors of cubs either at a similar time of day to the other videos or when near (but not in direct contact or even in the same habitat as) keepers? Or, could you have included non-petting and non-handling videos and compared them both within (behaviors during petting vs non-petting at same facility) and between facilities (behaviors during non-petting at different facilities)? Alternatively, could you have contacted some “real sanctuaries” and given them a survey that would indicate whether certain husbandry practices were used and cub behaviors were seen at that facility?

Line 115. Based on the number of users (44) vs number of videos (49), could/did some of the videos from the same user “double-dip” the same facility or even the same trip? Consider if two videos from the same trip at the same facility could still be considered independent observations. Similarly, were you able to tell the difference between individual facilities and determine the total number of different facilities at which footage was captured? Were any facilities sampled/recorded on video more than once or much more than other facilities?

Line 118-124. Can you please provide a little more information on how data were collected? Specifically, how behaviors were scored? From the blurb provided and some later text, it seems like all behaviors were scored by 1-0 sampling—either the behavior was seen or it wasn’t. Was this the method used? Or were you able to count and record the frequency (and duration, if applicable) of each behavior in each video? If you’ve only used 1-0, you should consider going back and looking at frequency and duration as a measure of intensity of these behaviors.

Line 127. From reading ahead, it looks like there were just 10 suspected volunteer tourism videos. While 2-sided t-test can be run with unequal sample sizes, did this lopsided distribution affect your variances between samples and/or did you have to use Welch’s t-test?

Line 128. To clarify: Does “total number of stress behaviors” mean total number of the 4 stress behaviors on your ethogram that occurred at least once, or does it mean the total frequency of these behaviors?

Line 128. What was the breakdown of videos across years? Did you have any videos from different years but at the same facility (i.e., to establish whether those facilities with poor conditions also had more observable stress behaviors)?

Line 131. Did you test for any correlations between number of husbandry practices seen and number of stress behaviors recorded?

Table 1. What is the functional difference between avoidance and hide behaviors? Their descriptions are similar, but I see in the results section that hide was rarely scored.

RESULTS

Line 158. Is “rate” the appropriate word here?

Line 165. I’ve never seen multiple p values reported as “p’s”

Figure 2. Although the axis for the Stress Behavior can go up to 5, there are only 4 stress behaviors on the ethogram and so only a max of 4 behaviors that could be reported, correct?

Figure 3. It’s not called out explicitly, but I’m assuming the probability in the (b) part of this figure is based on how many videos in each year included the behavior? For example, if there were 10 videos from 2008 and white lions were seen in 8 of them, the probability would be 80%--am I understanding this calculation correctly? Either way, you may want to explain how probability was determined.

Line 187-190, 195. If you didn’t have enough power to test individual behaviors, from where are you getting these p-values?

DISCUSSION

Line 205-207. Be careful here. Without baseline data for comparison (e.g., did a similar trend of increasing stress behaviors also occur in real sanctuaries?) or without a correlation run between behaviors and husbandry, it stretches a little to claim that an increase in stress behaviors were due to the inherent nature of the cub petting programs. Given that the increase happened while poor husbandry practices decreased and with no correlation to show a relationship either way, the (admittedly bad faith by way of example) claim could be made instead that it was the decrease of these poor husbandry behaviors that led to more stress behaviors. You need to either gather and include more statistical evidence or be slightly more careful in your wording for this section.

Line 218-220. This explanation of the differences between play and aggression should go in Methods, somewhere in the part where you introduce your ethogram.

Line 221. This sentence doesn’t seem exactly grammatically correct. Try: “Captive bred lions are farmed, and, much like domestic farm animals, they are handled by a variety…” or “Much like domestic farm animals, lions are also in effect farmed and handled by a variety…”

Line 223-226. Though it seems obvious and evident, do you have additional references investigating the effects of sleep disruption on cubs?

Line 247-249. These sentences are a bit redundant given the information in the above sentences. In the interests of conciseness, I suggest cutting these.

Line 250-254. Cut this paragraph. I understand the persuasive importance of including it, but the text starts to veer into speculation for these reasons: 1. Your work in the paper is focused on cub welfare and did not include a component on mother welfare, 2. You even state mother welfare is not sufficiently studied, so you shouldn’t make claims about suffering without proper referencing, 3. Don’t use a non-scientific documentary as a source, especially not as an anecdotal source.

Line 264-275, 279-314. Cut this section. It’s thematically related to your project, but detailed information about canned hunts, calls for awareness campaigns, and reminders about the harms of animal tourism in general are beyond the scope of the paper and the observations you performed. However, I did think lines 276-278 were an excellent and concise summary of your findings.

Lines 324-326. Please try to avoid these multi-line quotes.

CONCLUSION

Lines 331-348. Again, this information is beyond the scope of your paper. You can cut the conclusions down to just the end of line 348 - 355.

Reviewer 2 Report

This article gives an interesting insight in to welfare problems associated with captive lions housed at tourism facilities in South Africa. This is an under-researched, but important topic. The authors give a broad, well referenced overview of the industry in their introduction, and use a simple yet effective approach to gather and present their data. Their discussion details the potential problems associated with the lion cub petting industry in South Africa, based on their key findings put in the context of relevant literature surrounding this topic. Overall, I did not find any major issues or problematic sections that require extensive rewriting or revision. I have provided some suggestions for authors to consider adding or minor amendments that may improve the manuscript.

Specific comments for the submitting author to consider:

 (58) It may be more comprehensive to give the full range of reported number of lions and facilities across the key literature (although one citation is included in the manuscript, there are several more recent papers detailing different potential numbers). Notably –

  • Coals, P.; Burnham, D.; Loveridge, A.; Macdonald, D.W.; Sas-Rolfes, M.; Williams, V.L.; Vucetich, J.A. The Ethics of Human–Animal Relationships and Public Discourse: A Case Study of Lions Bred for Their Bones. Animals 2019, 9, 52.

  • Hutchinson, A.; Roberts, D.L. Differentiating Captive andWild African Lion (Panthera leo) Populations in South Africa, Using Stable Carbon and Nitrogen Isotope Analysis. Conserv. 2020, 29, 2255–2273.

(67) Would benefit from distinguishing between WTAs (which do not always involve direct contact with animals) and AVIs (animal visitor interactions). See –

  • D’Cruze, N., Khan, S., Carder, G., Megson, D., Coulthard, E., Norrey, J. and Groves, G., 2019. A global review of animal–visitor interactions in modern zoos and aquariums and their implications for wild animal welfare. Animals, 9(6), p.332.

(81) Consider whether the lion skeleton trade is best described as “illicit” – it is not forbidden in SA yet (although was declared constitutionally invalid in 2019).

(109) Would be beneficial so specify which “other resources” you are referring to

(114) Would be beneficial to confirm what experience the researcher identifying the age of the lions has. E.g. Was it a veterinarian or someone with extensive experience with captive lions?

(115) How many videos were there initially before you applied your exclusion criteria?

(Table 1) This table took a little time to get my head around and is quite confusing to the reader at first glance. Consider adding the column title “Categories” (to the far left column) and listing the three rows in this column ‘husbandry concerns, ‘stress behaviours’ and ‘tourism types’ (visiting tourist versus volunteer). OR consider breaking the table up in to separate tables, which would allow you to add more detail to the ethogram of stress behaviours. I think the description for the stress behaviours does need more detail, regardless of how you present the table. For example – for ‘hiding’ was there a set distance from humans cubs reached to classify as hiding, or was it a general attempt to move in the opposite direction to the humans? How long did they have to do it for, for the behaviour to count? Etc…). Some of the ‘variables’ would also benefit from more detail – for example, instead of “under 3 mo” consider “cubs younger than 3 months old present without a lioness”. This will really help clarity for the reader.

(Figure 2) The figure labels would be more clear to the reader if you changed them from ‘regular vs volunteer’ to ‘visiting tourist vs volunteer tourist’ - for clarity

(241) You could also add in here the nutritional deficiencies associated with mother cub separation

e.g. “Furthermore, cub separation from their mothers and the provision of alternative milk formulas (a practice reported at some lion farms [111]] can lead to nutritional deficiencies [129], which weakens immune systems and leaves animals more susceptible to pathogens [130].” From: Green, J., Jakins, C., Asfaw, E., Bruschi, N., Parker, A., de Waal, L. and D’Cruze, N., 2020. African lions and zoonotic diseases: implications for commercial lion farms in South Africa. Animals, 10(9), p.1692.

(252) The use of the term ‘crying out’ feels anthropomorphic and reduces the credibility of this paragraph. You have a strong enough point by saying “show cubs and mothers…suffering in myriad physical (e.g., bone 252 deformations and more genetic issues due to inbreeding and crossbreeding) and emotional (e.g., stress vocalizations) 253 ways” without using the term “crying for each other” first.

(256 – 262) This section may also benefit from reference to the quality of the enclosure (e.g. substrate, enrichment etc..) in addition to spatial considerations

(266) Could add that as well as educational campaigns for tourists, the industry would also benefit from increased transparency from facilities themselves – less green washing and misleading language, and from more regulation about what type of facilities can call themselves sanctuaries (sharing the responsibility with the tourists themselves)

(285 – 287) The relevance of placing this quote here is not entirely clear to me. If you are making the point that cubs are commercially driven/exploited because they are financially valuable this would benefit from an extra sentence or so clarifying this.

(341) This sentence would benefit from including a citation to a paper about zoonotic disease transmission from contact with wild animals (not lion specific, just the risk in general, to back up your point).

(354) While I understand the sentiment here, and agree with you, I think “respect for animals” can be subjective, and your conclusion would be stronger if you stick with reiterating that real sanctuaries do not allow direct contact between cubs and people (both tourists and volunteers alike

(General) You might also consider weaving into the discussion that these videos are only the ones deemed ‘acceptable/appropriate’ to post by the tourists and volunteers, and they still show stress and poor welfare – so there may be many more interactions that have far worse welfare that are more overtly ‘bad’ or ‘cruel’ at these types of facilities that are never posted online

(General) I think it is necessary to acknowledge as a limitation that some of the information may not be reliable as it was taken from the video descriptions (e.g. the age of the cubs interacting with people)